# The Regulatory Network of Sweet Corn (*Zea mays* L.) Seedlings under Heat Stress Revealed by Transcriptome and Metabolome Analysis

**DOI:** 10.3390/ijms241310845

**Published:** 2023-06-29

**Authors:** Zhuqing Wang, Yang Xiao, Hailong Chang, Shengren Sun, Jianqiang Wang, Qinggan Liang, Qingdan Wu, Jiantao Wu, Yuanxia Qin, Junlv Chen, Gang Wang, Qinnan Wang

**Affiliations:** 1Institude of Nanfan & Seed Industry, Guangdong Academy of Sciences, Guangzhou 510640, China; 2Hainan Yazhou Bay Seed Laboratory, Sanya 572024, China; 3College of Horticulture and Landscape Architecture, Southwest University, Chongqing 400715, China

**Keywords:** sweet corn, heat stress, transcriptomics, metabolomics, metabolites

## Abstract

Heat stress is an increasingly significant abiotic stress factor affecting crop yield and quality. This study aims to uncover the regulatory mechanism of sweet corn response to heat stress by integrating transcriptome and metabolome analyses of seedlings exposed to normal (25 °C) or high temperature (42 °C). The transcriptome results revealed numerous pathways affected by heat stress, especially those related to phenylpropanoid processes and photosynthesis, with 102 and 107 differentially expressed genes (DEGs) identified, respectively, and mostly down-regulated in expression. The metabolome results showed that 12 or 24 h of heat stress significantly affected the abundance of metabolites, with 61 metabolites detected after 12 h and 111 after 24 h, of which 42 metabolites were detected at both time points, including various alkaloids and flavonoids. Scopoletin-7-o-glucoside (scopolin), 3-indolepropionic acid, acetryptine, 5,7-dihydroxy-3′,4′,5′-trimethoxyflavone, and 5,6,7,4′-tetramethoxyflavanone expression levels were mostly up-regulated. A regulatory network was built by analyzing the correlations between gene modules and metabolites, and four hub genes in sweet corn seedlings under heat stress were identified: *RNA-dependent RNA polymerase 2* (*RDR2*), *UDP-glucosyltransferase 73C5* (*UGT73C5*), *LOC103633555*, and *CTC-interacting domain 7* (*CID7*). These results provide a foundation for improving sweet corn development through biological intervention or genome-level modulation.

## 1. Introduction

In nature, plants face a range of biotic and abiotic stressors, including heat, drought, and waterlogging. However, the negative impacts of high temperatures are particularly pronounced. Greenhouse gas emissions, such as CO_2_, have accelerated the occurrence of extreme climates, leading to a global temperature rise of over 1 °C in the past 50 years [1,2]. High temperature is a major threat to crop productivity, negatively impacting plant development and yield [3,4,5,6]. Climate change has significantly impacted global food production, with wheat and rice yields declining by 5.0 and 1.6 million tons annually, respectively [7]. A multi-ensemble model predicted that between 1981 and 2010, maize and soybean yields declined by 4.1% and 4.5%, respectively, even when considering the effects of CO_2_ fertilization and modernized agronomic practices [8]. The total factor productivity (TFP) econometric model suggests that anthropogenic climate change, especially heat stress, has made global agriculture more vulnerable to ongoing climate change [9]. Plants may experience heat waves during various developmental stages, including the seedling, vegetative, and reproductive stages, which can pose significant challenges to growth, reproduction, and other developmental processes [3].

Plants have evolved multiple physiological, molecular, and metabolic mechanisms to perceive, transduce, and respond to heat stress throughout their growth cycle [4,5]. Heat stress-induced fluidity of the plasma membrane triggers downstream signaling events, including Ca^2+^ signaling, lipid signaling, the mitogen-activated protein kinase (MAPK) signaling cascade, respiratory burst oxidase homologs (RBOH)-dependent reactive oxygen species (ROS) signaling, and a series of stress-responsive gene expressions to promote plant adaptation to adverse environments [10,11,12,13]. In plants, the classical heat stress response models involve the induction of heat shock proteins (HSPs) and the regulation of heat shock factors (HSFs) in response to protein denaturation [14]. While the discovery of heat stress sensors in the plasma membrane, which could be responsible for the initiation of plant heat stress response (HSR), suggests the presence of other pathways during HSR [5]. For the HSFs-dependent heat response, HSFA1s are the master regulators and manipulate diverse transcription factors (TFs), signaling molecules, such as *HSFB2*, *dehydration-responsive element-binding 2A* (*DREB2A*), calcium signaling, ROS signaling, etc., to induce the expression of downstream genes [5,14]. To improve plant heat adaptation, studies have verified the thermal tolerance of many genes—for example, *TaHsfA6f*, *OsHSP18.6*, *ZmMAPK1*, and *ZmWRKY106*—using transgenic methods, and results indicated that those candidate genes have big potential [3]. In rice, the transduction of heat signaling from the plasma membrane to the chloroplast depends on the quantitative locus *Thermo-tolerance 3* (*TT3*), which includes two genes, *TT3.1* and *TT3.2* [15].

Phytohormones also play an important role in plant response to heat stress [16]. For instance, the *phytochrome-interacting factor 4* (*PIF4*)/*PIF7*-mediated thermomorphogenesis depends on the regulation of auxin biosynthesis genes, which results in cell elongation in petioles and hypocotyls by activating brassinosteroid (BR) biosynthesis and signaling [17]. Moreover, the plasma membrane signaling stimulated by increased temperature may perceive and transduce through TOT3-mediated brassinazole-resistant 1 (BZR1) activity to regulate plant thermomorphogenesis [18]. Mutants from abscisic acid signaling (*abi1* and *abi2*), ultraviolet (UV)-sensitive (*uvh6*), ROS (*atrbohB* and *D*), and salicylic acid (SA) biosynthesis (*NahG*) have shown defects in acquired thermotolerance of root growth and seedling survival, indicating the potential role of abscisic acid (ABA), SA, ROS, and UV in plant thermotolerance [19]. Identifying key genes or loci is necessary to develop new strategies for breeding heat-resistant crops.

In addition to genes, metabolites produced during primary and secondary metabolism are also involved in the complex processes triggered by heat stress in plants [20]. The biosynthesis of these molecules is considered a powerful strategy for plants to cope with stressful growth conditions. Reports have shown that the production of polyamines and subsequent formation of phenylamides are necessary for ROS scavenging during heat shock and water stress [21,22]. A wide range of terpenoids, phenolics, flavonoids, and alkaloids have also been reported to be involved in coping with stress and defensive stimuli during plant growth [20,23]. In sweet corn, response to extreme temperatures after germination is not only mediated by phytohormones, but the changing of volatiles also appears necessary, particularly methyl linoleate, 2,3-butanediol, and 2,3-butanedione [24]. In the hybrid maize ZD309, plant hormone signal transduction, cysteine and methionine metabolism, and α-linolenic acid metabolism play crucial roles in heat tolerance [25]. The maize NAC transcription factor ZmNAC074 increases heat stress tolerance by modulating the abundance of stress metabolites, such as ROS, antioxidants, malondialdehyde (MDA), proline, soluble protein, chlorophyll, and carotenoids [26]. Although genes and metabolites synergistically affect the plant stress response, the application of metabolites in maize agriculture is limited. Mining the internal connections between metabolites and genes may provide more potential for cultivating resistant crops.

Sweet corn (*Zea mays* L.) is a popular food worldwide due to its wonderful flavor and is suggested to be a functional food due to its abundance of essential nutrients for humans [27]. However, rising temperatures surpassing the upper-temperature threshold during the anthesis stage have resulted in a reduced yield of 0.5–2% over 27 years in the United States [28]. It is, therefore, urgent to accelerate sweet corn adaptation to sustain its production under adverse circumstances. This study was undertaken to illustrate the regulatory network in sweet corn at the seedling stage, based on the differentially expressed genes (DEGs) and their participating pathways detected, integrated with the in vivo metabolic changes during heat stress.

## 2. Results

### 2.1. RNA Sequencing Results and DEGs Detected after Heat Stress

A total of 103.16 GB of clean data was obtained from the 15 samples. Each library yielded more than 6 GB of clean data, with about 92% Q30 bases and a GC content of around 55%. Between 82.03% and 86.32% of unique, clean reads were mapped to the reference genome GCF_902167145.1_Zm-B73-REFERENCE-NAM-5.0_genomic (Appendix A). The correlation coefficient values among the samples showed good repeatability within the same group (Appendix A), which was also confirmed by principal component analysis (PCA) (Figure 1A).

Various physiological and biochemical processes are activated in plants under heat stress. To investigate the relationship between gene expression and heat stress, we identified 7800 DEGs with the selection criteria of |Log_2_FoldChange (Log_2_FC)| ≥ 1 and false discovery rate (FDR) < 0.05 (Appendix A). Samples with and without heat stress were clearly separated based on the expression heatmap (Appendix A). The DEGs were differentially expressed at all time points when sweet corn plants were under heat stress. The volcano plot showed an apparent increase in the number of DEGs with stress duration, especially for up-regulated genes (Figure 1C,D). Between CK12h and HT12h, there were 3641 DEGs, of which 1638 were up-regulated, and 2003 were down-regulated. The number of DEGs increased to 5934 between CK24h and HT24h, of which 3129 were up-regulated, and 2805 were down-regulated. Additionally, 1775 DEGs were detected at both time points for CK vs. HT (Figure 1B).

### 2.2. KEGG and GO Analysis for DEGs

All DEGs were classified into five KEGG categories, which were distributed across 137 metabolic pathways. KEGG enrichment analysis revealed that 33 metabolic pathways were significantly enriched under heat stress, and the 20 most enriched pathways were shown in Figure 1E,F for different time points. Detailed information is listed in Appendix A. At 12 h after heat stress, the five most enriched pathways were biosynthesis of secondary metabolites, phenylpropanoid biosynthesis, plant–pathogen interaction, alpha linolenic acid metabolism, photosynthesis–antenna proteins, and MAPK signaling pathway. At 24 h, the most enriched pathways were biosynthesis of secondary metabolites, phenylpropanoid biosynthesis, starch and sucrose metabolism, circadian rhythm plant, carotenoid biosynthesis, and photosynthesis–antenna. Notably, biosynthesis of secondary metabolites, phenylpropanoid biosynthesis, and photosynthesis–antenna proteins were detected after both short- and long-term heat stress.

For GO analysis, 91 and 82 DEGs were annotated for CK12h vs. HT12h and CK24h vs. HT24h, respectively. Appendix A show the top 50 items, with 39 and 29 GO items annotated in the biological process part between the two groups, 10 and 17 GO items marked in the molecular function part, and only 1 and 3 GO items marked in the cellular component part (Appendix A for details). The six most enriched items in the CK12h vs. HT12h group were RNA modification, chloroplast RNA modification, phenylpropanoid metabolic process, protein folding, response to heat, and secondary active transmembrane transporter activity. In contrast, the most enriched items in the CK24h vs. HT24h group were response to hydrogen peroxide, monooxygenase activity, response to an antibiotic, cellular response to hypoxia, response to hypoxia, and cellular response to decreased oxygen levels. Sixteen GO items were detected in both groups, and the biological process phenylpropanoid metabolic process and response to heat appeared to be more important based on the *p*-values.

In conclusion, DEGs in numerous pathways were involved in the heat stress response, and the phenylpropanoid metabolic process may be an important pathway involved in sweet corn thermotolerance.

### 2.3. Heat Stress Stimulates Abundance Variation of DEGs in Phenylpropanoid Biosynthesis and the Phenylpropanoid Metabolic Process

Phenylpropanoid compounds and their derivatives encompass various structural classes with diverse biological and physiological functions, including coumarin derivatives, phenylpropanoids, isoflavones, lignin, flavones, spermidine derivatives, flavonols, tannins, and anthocyanins. These compounds are tightly associated with normal plant development and responses to changing environmental stimuli [29].

This study investigated the processes of phenylpropanoid biosynthesis and phenylpropanoid metabolism, with 102 DEGs encoding 20 kinds of bio-enzymes (Figure 2A and Appendix A). After 24 h of heat stress, the genes encoding enzymes in the core phenylpropanoid biosynthesis pathway were all inhibited, including *PAL* (*phenylalanine ammonia-lyase*, ten genes), *C4H* (*cinnamic acid 4-hydroxylase*, three genes), and *4CL* (*4-coumarate-CoA ligase*, three genes). In addition, genes encoding *cinnamyl alcohol dehydrogenase* (*CAD*, six genes, down-regulated), *peroxidase* (*PER*/*POD*, 33 genes, five up-regulated, 28 down-regulated), *cinnamoyl-CoA reductase* (*CCR*, eight genes, four up-regulated, four down-regulated), *trans-caffeoyl-CoA 3-O-methyltransferase* (*COMT*, two genes, down-regulated), *ferulate-5-hydroxylase* (*F5H*, two genes, down-regulated), *cytochrome P450 98A1* (*CYP98A*, one gene, down-regulated), and *putrescine hydroxycinnamoyl transferase*/*hydroxycinnamoyl transferase-encoding genes* (*PHT*/*HCT*, six genes, one up-regulated, five down-regulated), as well as *caffeoyl shikimate esterase* (*CSE*, five genes, one up-regulated, four down-regulated), which are directly involved in the monolignol biosynthesis, were mostly down-regulated.

Two *UDP-glycosyltransferase-encoding* genes involved in scopolin biosynthesis (*SCGT*, two genes, down-regulated), ten DEGs involved in the coumarin biosynthesis pathway (*BGL*, five up-regulated, five down-regulated), four *2-hydroxyisoflavanone dehydratase-encoding* genes related to isoflavone biosynthesis (*HIDM*, one up-regulated, three down-regulated), one gene encoding phenylcoumaran benzylic ether reductase (*PYRC5*), four genes related to fatty acids (*CYP86B1*, *AL2C4*, and *HHT1*), and two wall lignin-modifying (*MWL1*) genes were down-regulated (Appendix A). These results suggest that high temperature decreases monolignol biosynthesis and loosening of the cell wall, promoting the fluidity of the plasma membrane to facilitate the influx of substances, biomass, and information.

The RT-qPCR results of 34 candidate genes showed a similar expression pattern to those of RNA-seq files based on the fold changes of most DEGs (Figure 2B), further validating the findings from our transcriptome study. Details of the results are shown in Appendix A.

### 2.4. Extreme High Temperature Seriously Affects Photosynthesis in Sweet Corn

Chlorosis only occurred on the blade tip of the first leaf after 12 h of heat stress. However, as the heat continued for 24 h, the first leaf became dry and yellowish, and the growth of the whole plant was significantly restrained (Appendix A). To confirm the recovery capability of Wantian 2015 after 12 and 24 h of heat treatment, we moved the plants to 25 °C for another 24 h and measured indexes related to photosynthesis, including maximum electron transport rate (ETR_max_), saturating irradiance (E_K_), maximum photosynthetic rate (F_v_/F_m_), actual photosynthetic rate (Y(II)), photochemical quenching coefficient (qP), and quantum yield of regulatory energy dissipation (Y(NPQ)).

As shown in Figure 3C, after 24 h of recovery, the seedlings that had suffered heat stress increased the ETR_max_, Y(II), and qP values from 49.82 ± 2.528, 0.0213 ± 0.001, 0.0389 ± 0.003 at 25 °C to 61.62 ± 2.669, 0.032 ± 0.001, and 0.064 ± 0.002 at 42 °C. Meanwhile, we also compared the values of F_v_/F_m_, E_K_, and Y(NPQ) between the two groups, and no significant difference was detected (with *q* values > 0.05 for all). These results indicate that the photosynthesis of Wantian 2015 could recover to a normal state soon after the heat stress disappeared, with more efficient photosynthesis and less dissipation. This may be the reason why Wantian 2015 has a better adaptation to climatic variations.

Our RNA-seq data revealed a total of 107 DEGs related to photosynthesis. Under heat stress, the antenna protein genes *Lhca* and *Lhcb*, which encode the light-harvesting chlorophyll protein complex (LHC), were down-regulated, except for *LOC100193833* (*Lhcb1*), *LOC100281248* (*Lhcb1*), and *LOC542478* (*Lhcb4*), which showed opposite expression trends under 12 h of heat stress but had decreased expression or no significant variation later (Figure 3A and Appendix A).

Heat stress not only reduced sweet corn’s light capture capability but also inhibited the expression of genes involved in photosystems I (*Psa*, all nine DEGs were down-regulated) and II (*Psb*, nine DEGs were up-regulated, and 11 DEGs were down-regulated) reaction center proteins or subunits (Figure 3B and Appendix A), indicating a decrease in the photosynthetic rate. Furthermore, heat stress affected genes encoding the cytochrome b6/f complex and photosynthetic electron transport, including one *PetA*, eleven *PetF*, and four *PetH*. The expression of *PetA* was up-regulated, while that of seven *PetF* and two *PetH* genes was down-regulated, indicating a relatively lower energy conversion efficiency when sweet corn is under heat stress.

Nineteen photosynthetic carbon fixation genes showed decreased expression, while 12 other genes were up-regulated (Figure 3G and Appendix A). Heat stress also affected genes involved in chlorophyll biosynthesis and catabolism, photosystem II assembly and repair, and regulation of the photosynthesis process (Figure 3D–F and Appendix A).

In summary, heat stress inhibited various physiological and biochemical processes in chloroplasts, including photosynthetic pigments, photosystems, primary reaction, photosynthetic electron transport, and carbon fixation.

### 2.5. Metabolic Changes in Sweet Corn Leaves under Heat Stress

UPLC-ESI-MS/MS identified 819 distinctive metabolites in ten categories under heat stress. These included flavonoids, alkaloids, amino acids and derivatives, phenolic acids, lipids, nucleotides and derivatives, organic acids, lignans and coumarins, terpenoids, and others (Appendix A and Appendix A). As expected, PCA analysis showed a distinctive separation between the heat-treated and control groups. The PC1 score was 24.46%, and the PC2 score was 14.18%, indicating that short or long heat stress significantly impacted metabolism (Appendix A). The predicted PCA by OPLS-DA between the two time points is shown in Appendix A, where the T score was 33.7% and 38.9%, and the orthogonal T score was 20.8% and 19.7% for the groups CK12h vs. HT12h and CK24h vs. HT24h, respectively. The accuracy of the OPLS-DA results was verified by 200 random permutations and combined alignment tests. The *p*-values for Q^2^ and R^2^Y for both groups were lower than 0.05, indicating the reliability of the OPLS-DA model (Appendix A).

We identified differential metabolites from the two groups based on their variable importance in projection (VIP) from OPLS-DA. The fold change from PCA was used to further screen the compounds. A total of 61 and 111 differential metabolites were identified between the groups CK12h vs. HT12h and CK24h vs. HT24h, respectively, according to the standard VIP ≥ 1 and |Log_2_FC| ≥ 1. These included 28 flavonoids, 24 alkaloids, 23 amino acids and derivatives, 18 phenolic acids, 13 nucleotides and derivatives, 10 lipids, 8 organic acids, 3 lignans and coumarins, and 1 terpenoid, among others.

Compared to the control group, the metabolites that mostly increased in abundance after 12 h of heat stress were cryptochlorogenic acid (4-o-caffeoylquinic acid), scopoletin-7-o-glucoside (scopolin), 3-indolepropionic acid, 5,7-dihydroxy-3′,4′,5′-trimethoxyflavone, quercetin-3,4′-o-di-glucoside, 5,6,7,4′-tetramethoxyflavanone, and *p*-coumaroyltyramine, while dihydroxy-dimethoxyflavone, kaempferol-3-o-(2-o-xylosyl-6-o-rhamnosyl) glucoside, 3′-adenylic acid, (S)-2-hydroxy-3-(4-hydroxyphenyl) propanoic acid*, and 2-deoxyribose-1-phosphate had mostly decreased abundance (Figure 4A and Appendix A). After 24 h of heat stress, the mostly accumulated metabolites were scopoletin-7-o-glucoside (scopolin), petroselinic acid, 1-methyladenine, 5,7-dihydroxy-3′,4′,5′-trimethoxyflavone, 5,6,7,4′-tetramethoxyflavanone, ursolic acid*, 3-ureidopropionic acid, *p*-coumaroyltyramine, l-phenylalanine, and 1-o-feruloyl-*β*-d-glucose, while dihydroxy-dimethoxyflavone, 2-caffeoyl-l-tartaric acid (caftaric acid), 6′-o-feruloyl-d-sucrose, taxifolin (dihydroquercetin), *n*-caffeoylputrescine, dopamine, and 1-o-p-coumaroylquinic acid were the mostly decreased metabolites (Figure 4B and Appendix A).

A total of 130 significant compounds were clustered into five classes in a timely manner, as depicted in Figure 4C and listed in Appendix A. In response to heat stress, 42 metabolic compounds, including 12 amino acids and derivatives, 11 alkaloids, 6 flavonoids, 5 phenolic acids, 4 nucleotides and derivatives, 2 organic acids, 1 coumarin, and 1 lipid, were detected after both 12 and 24 h of 42 °C heat stress. The expression trends of these compounds are shown by the heatmap in Figure 4D.

Scopoletin-7-o-glucoside (scopolin), 3-indolepropionic acid, acetryptine, 5,7-dihydroxy-3′,4′,5′-trimethoxyflavone, 5,6,7,4′-tetramethoxyflavanone, and taxifolin (dihydroquercetin) were the six metabolites with the highest abundance, classified into lignans and coumarins (one compound), alkaloids (two compounds), and flavonoids (three compounds). Except for taxifolin (dihydroquercetin), flavonoids (Figure 4C, class 2; Figure 4D, number 34 in Appendix A) showed a downward trend during heat stress. 5,7-dihydroxy-3′,4′,5′-trimethoxyflavone, 5,6,7,4′-tetramethoxyflavanone, and scopoletin-7-o-glucoside (scopolin) were barely detected in the control group, but heat stress dramatically increased their relative abundance (Figure 4C, classes 1 and 4; Figure 4D, numbers 36, 37, and 38 in Appendix A).

3-indolepropionic acid and acetryptine were rarely detected in the control at 12 h but accumulated significantly at 24 h, indicating their specific functional role during sweet corn growth. After 12 h of heat stress, 3-indolepropionic acid and acetryptine had tremendous accumulation, with Log_2_FC values of 11.8 and 10.4, respectively (Figure 4C, class 4; Figure 4D, numbers 25 and 26 in Appendix A), indicating their specific roles not only in plant growth but also in abiotic stress response.

### 2.6. Correlations among Plant Metabolites under Heat Stress in Sweet Corn

To better understand the inter-regulation of metabolites during heat stress in sweet corn, we selected 14 metabolites with the most easily detectable abundance for correlation evaluation based on their VIP and FC values. The results of the Spearman correlation analysis are shown in Figure 4E and Appendix A. We found a positive correlation value of 1 (*p* < 0.001) between 5,7-dihydroxy-3′,4′,5′-trimethoxyflavone and 5,6,7,4′-tetramethoxyflavanone. Alkaloids, 3-indolepropionic acid, and acetryptine were also positively correlated with 5,7-dihydroxy-3′,4′,5′-trimethoxyflavone and 5,6,7,4′-tetramethoxyflavanone, with a correlation value of 0.86 (*p* < 0.001). Phenolic acid, 2-caffeoyl-l-tartaric acid (caftaric acid), acted antagonistically to alkaloids, 3-indolepropionic acid, and acetryptine (correlation value −0.94 and −0.91, *p* < 0.001), as well as to flavonoids 5,7-dihydroxy-3′,4′,5′-trimethoxyflavone and 5,6,7,4′-tetramethoxyflavanone (correlation value −0.92, *p* < 0.001). Another alkaloid, dopamine, showed an inverse correlation with 5,7-dihydroxy-3′,4′,5′-trimethoxyflavone and 5,6,7,4′-tetramethoxyflavanone (−0.93, *p* < 0.001).

### 2.7. Detection of Correlations among Gene Modules and Metabolite Traits via WGCNA Analysis

To reveal the relationships between genes and metabolites and identify new pathways contributing to the construction of metabolic networks, 19 metabolites were selected at 12 h or 24 h after heat stress for weighted gene co-expression network analysis (WGCNA). Higher correlations were found between modules and several compounds (Figure 5A). Four modules were selected for KEGG enrichment analysis (Figure 5B–E). 

The brown module had high correlation values with 1-o-feruloyl-*β*-d-glucose, kaempferol-7-o-rhamnoside, and anthranilate-1-o-sophoroside with values of 0.87, 0.86, and 0.85, respectively (Figure 5A). The second and third most enriched pathways were arginine biosynthesis and glutathione metabolism, respectively (Figure 5B), indicating potential links between metabolites and these pathways.

The dark green module was correlated to 2-deoxyribose-1-phosphate with a value of 0.85 (Figure 5A). 2-deoxyribose-1-phosphate belongs to the pentose phosphate pathway (PPP), which is an essential pathway that provides tryptophan precursors. The most enriched pathway was tryptophan biosynthesis in Figure 5C, providing evidence for gene regulation of 2-deoxyribose-1-phosphate reduction of sweet corn seedlings under heat stress. 

The magenta module had a close relationship with 5,7-dihydroxy-3′,4′,5′-trimethoxyflavone with a correlation value of 0.89 (Figure 5A). KEGG enrichment analysis showed that genes were processed in the endoplasmic reticulum, ribosome, and endocytosis (Figure 5D), supporting the model that flavonoid metabolism is catalyzed by an enzyme complex localized to the endoplasmic reticulum, which may also exist in sweet corn [30]. 

The tan module was only correlated with taxifolin (dihydroquercetin) (Figure 5A). The second KEGG classification group was cofactors biosynthesis (Figure 5E), indicating that a decreased amount of taxifolin (dihydroquercetin) might be needed to regulate cofactors. 

To identify the potential key genes involved in regulating the complex networks under heat stress, we gathered the hub genes for each module according to their connectivity (also called degree). The hub genes for the brown, dark green, magenta, and tan modules were *LOC100502460* (*RNA-dependent RNA polymerase 2*/*SILENCING MOVEMENT DEFICIENT 1*, *RDR2*/*SMD1*), *LOC100285624* (*UDP-glucosyltransferase 73C5*, *UGT73C5*), *LOC103633555*, and *LOC100281580* (*CTC-interacting domain 7*, *CID7*), respectively. Potentially connected genes were selected with the Log_2_FC ≥ 1.0 for *UGT73C5* and Log_2_FC ≥ 1.5 for *RDR2*, *LOC103633555*, and *CID7*; the weight ≥0.10 for each hub gene. 

Two-hundred-forty-eight up-regulated DEGs were possibly connected to the hub gene *RDR2* in the brown module (Figure 5F). The most connected genes were *ACOX2* (*Acyl-coenzyme A oxidase 2*) and *TBL29* (*trichome birefringence-like 29*), which had weight values of 0.324 and 0.323 (Appendix A). Only 12 DEGs were related to *UGT73C5* in the dark green module, and the most correlated gene, *C92C5* (*Cytochrome P450 92C5*), was dramatically down-regulated, with the Log_2_FC value as −3.844 (Figure 5G; Appendix A). The hub gene *LOC103633555* from the magenta module was connected to 140 up-regulated DEGs. The genes *novel 2788* and *HSP7C* (*heat shock cognate protein 7*) owned the most weight at the same value of 0.33 (Figure 5H; Appendix A). The hub gene *CID7* connected to 23 down-regulated DEGs in the tan module with the most tightly related genes *LPCT2* (*lysophospholipid acyltransferase 2*) and *EDR2L* (*enhanced disease resistance 2-like*) (Figure 5I; Appendix A). The genes with the largest weight for each module may also be the most important hub genes to respond to heat stress in sweet corn.

## 3. Discussion

Climate change caused by greenhouse gases has led to a large-scale reduction of crop productivity, resulting in frequent and extreme heatwaves and droughts in many regions. This has threatened global food security and curtailed biomass feedstock and biofuel production [2]. High temperatures have also hindered normal plant growth by disrupting the photosynthetic system, cell growth and division, and reproduction, among other processes [6,31,32]. It is urgent to accelerate crop improvement to better tolerate and adapt to climate change. Evidence shows rising temperatures expedite sweet corn yield losses [28]. However, comparatively little study has been conducted regarding the response mechanism. Therefore, it is urgent to accelerate crop adaptation strategies to sustain production. Although metabolic changes due to heat stress have been investigated in many plants, none have been undertaken in sweet corn. This study aims to uncover the regulatory network between genes and metabolites in sweet corn at the seedling stage under heat stress through combined transcriptomic and metabolomic analysis.

Secondary metabolites not only contribute to specific odors, flavors, and colors in plants but are also important for plants to interact with adverse environments for adaptation and defense [33]. In our study, transcriptomic data showed that dramatic changes in DEGs occurred and clustered to the secondary metabolite pathway, with 359 and 532 genes being down-regulated after 12 h and 24 h of heat stress, respectively. These genes accounted for 29.69% and 27.30% of the DEGs, respectively. Upon detailed analysis, genes from phenylpropanoid biosynthesis and photosynthesis–antenna proteins in the KEGG metabolism category also showed an extremely high enrichment with high rich factor and *p*-value after both 12 h and 24 h of heat stress (Figure 1E,F). Transcriptome analysis of maize inbred line B73 seedlings also indicated that biosynthetic pathways of secondary metabolites and protein synthesis in the endoplasmic reticulum might play a central role in maize response to heat stress [34], and those pathways were also enriched in our results. The hydrogen peroxidase metabolic process, cellular modified amino acid metabolic process, and sulfur compound metabolic process were common response pathways detected in maize seedlings in response to both heat and cold stress [35]. 

Nevertheless, Xiang et al. [24] reported that plant hormones and high temperatures were important in inhibiting the accumulation of zeatin, salicylic acid, jasmonic acid, and auxin. Phytohormones played a key role in the regulative network under heat stress at the early growing stage of sweet corn seedlings. Comparative transcriptome analysis also found the vital role of up-regulated genes in the photosynthesis process and indicated that chloroplast proteins might play an important role in increasing heat tolerance in sweet corn [36]. However, the DEGs involved in the photosynthesis process were mostly restrained in our study (Figure 3). The most significant metabolic pathway for up-regulated genes was the secondary metabolite biosynthetic pathway, which was contrary to the results of previous research. In addition to the difference in varieties, the duration of heat stress in the previous research was only 0.5 h and 3 h. In general, sweet corn may respond to heat waves by using a synergistic or antagonistic manner through activated identical or disparate metabolic pathways due to the varieties, genotypes, and the frequency and intensity of rising temperatures.

Phenylpropanoid secondary metabolites serve as antioxidants for efficient quenching of singlet oxygen radicals in plant cells during various stresses. However, fixed carbon through photosynthesis is the source of secondary metabolites with an overall effect on growth inhibition. Some synergistic effects may be encountered in plant systems [20]. In our study, heat stress impacted the physiological and biochemical processes in chloroplasts of sweet corn by affecting gene expression related to photosynthetic pigments, photosynthetic electron transport, carbon fixation, and more (Figure 3). We also detected significant variations of gene expression in regulating phenylpropanoid biosynthesis and metabolic pathways. Under high temperatures, genes involved in these processes were mostly down-regulated, such as *PALs*, *4CLs*, *PODs*, and *HCT*/*PHTs* (Figure 2). As one of the most important metabolic pathways, multiple branches of phenylpropanoid metabolism are involved in abiotic responses, especially lignin and flavonoids [37]. Intriguingly, the repression of lignin biosynthesis by *HCT* silencing leads to the redirection of metabolic flux to the biosynthesis of flavonoids [38]. Thus, the dramatic suppression of *HCTs* may account for the enrichment of flavonoid biosynthesis and the plentiful accumulation of compounds like 5,7-dihydroxy-3′,4′,5′-trimethoxyflavone, 5,6,7,4′-tetramethoxyflavanone, quercetin-3,4′-o-di-glucoside, and kaempferol-7-o-rhamnoside (Log_2_FC were 10.3479, 9.6234, 9.7919, and 2.0820 after 12 h of stress), based on our metabolomic data. This indicates that *HCTs* play a potential regulatory role not only in plant growth but also in response to heat stress. However, the down-regulation of lignin biosynthesis genes does not necessarily impair plant defenses, and in some cases, plants with reduced lignin have increased resistance to phytopathogens. Thus, a reduced number of lignin-related genes may also be necessary for heat response in sweet corn, which may explain the down-regulation of DEGs in the phenylpropanoid and lignin biosynthesis pathways in our data. Considering the inhibitory role of superfluous flavonoids to auxin transport and growth [38], another 11 flavonoids were down-regulated, especially taxifolin (dihydroquercetin), which had a Log_2_FC of −12.1 after 24 h of stress, to balance the possible severe consequences for sweet corn under heat stress. 

The accumulation of auxin precursors, such as 3-indolepropionic acid and acetryptine, may be another efficient strategy to balance growth and defense when plants suffer from a relatively long period of heat stress. This also demonstrates another key role of the auxin signaling network in defense response. Flavonoids and auxin may synergistically affect plant heat response, which has been verified by the positive correlation between 3-indolepropionic acid and acetryptine and 5,7-dihydroxy-3′,4′,5′-trimethoxyflavone and 5,6,7,4′-tetramethoxyflavanone (Figure 4E). Scopoletin and its glucosylated form, scopolin, have antimicrobial and antioxidative activities and appear to play an important role in disease resistance [29,39], but few reports have related these to abiotic stress. In our study, heat stress also promoted abundant production of scopolin, which could play antioxidative activities when plants encounter heat stress. In summary, we found a possible regulatory pathway among the metabolites related to phenylpropanoid, flavonoid, auxin, and scopolin and provided fundamental work to comprehend the intricate relationships in sweet corn seedlings under heat stress. Additional research is needed to elaborate on the concrete responsive mechanism.

Plants have developed sophisticated and subtle signaling networks to perceive climate change, induced relative signaling, and activated heat-responsive gene expressions during the long evolutionary period [5]. Mining the key genes during such molecular processes could facilitate the development of molecular markers, which are necessary to breed heat-tolerant crop species [5]. To uncover the possible molecular processes and the key genes contributing to thermotolerance, we conducted a WGCNA analysis between genes and metabolites. Four modules were selected according to the correlation values. Seven pathways were analyzed, including arginine biosynthesis, glutathione metabolism, tryptophan biosynthesis, endoplasmic reticulum, ribosome, endocytosis, and biosynthesis of cofactors (Figure 5). Arginine biosynthesis and tryptophan biosynthesis were involved in the heat response process at an early stage of seedlings in sweet corn [24]. KEGG pathway enrichment analysis for DEGs in B73 indicated that protein processing in the endoplasmic reticulum played a central role in maize response to heat stress [34], and short-term heat stress significantly inhibited gene expression in ribosomes [36]. These results coincide with ours, indicating the importance of arginine biosynthesis, tryptophan biosynthesis, endoplasmic reticulum, and ribosome in corn’s response to heat. However, the importance of glutathione metabolism, endocytosis, and biosynthesis of cofactors still needs more proof.

We detected four hub genes for each module, which may be potential heat-tolerance-related marker genes in sweet corn. RDR2/SMD1 has been suggested to attend plant flowering and play roles in broad resistance to *Meloidogyne* spp. in plants, although its regulation mechanism is still unknown [40,41]. In our study, as many as 248 up-regulated genes connected with this gene. Brassinosteroids (BR) provide heat tolerance via compensation and priming of genes. *UGT73C5* is involved in the homeostasis of steroid hormones to regulate BR activity [42,43]. *UGT73C5* may also contribute to heat stress tolerance in sweet corn through fine-tuning BR, but more research is needed for this. *LOC103633555* and *CID7* are noteworthy with a big potential because of *LOC103633555*’s tight connection with *HSPs* and *CID2*’s close relation with the genes *KEA2*, *LPCT2*, and *EDR2L* that are involved in abiotic stress response and plant development processes. Further studies are now needed to investigate the relationships between those genes and heat stress response in sweet corn.

## 4. Materials and Methods

### 4.1. Growing Conditions and Stress for Sweet Corn Seedings

Seeds of sweet corn variety Wantian 2015 were planted in a seedling tray and cultured in a growth chamber with 16 h light, 8 h dark, and 65% humidity. The nursery substrate was mixed with soil/perlite/vermiculite (3/1/1, *v*/*v*/*v*). Germinated seeds were cultivated with adequate irrigation and given water-soluble compound fertilizer once a week. After 21 days of growth to the trefoil stage, the fully extended and photosynthetically mature second leaves from five seedlings were taken for each sample after 42 °C for 12 h and 24 h, and control leaves for each time point were also taken at 25 °C, 0 h as blank control. Three biological replications were set for each time point. Samples for RNA-seq and metabolites were taken at the same time. A total of 15 samples were collected for each experiment, frozen quickly in liquid nitrogen, stored at −80 °C, and then shipped on dry ice to Metware Biotechnology Co., Ltd. (Wuhan, China) for RNA extraction, quality detection, and library construction.

### 4.2. Library Preparation, Sequencing, and Data Analysis

One microgram of RNA was used for each library. Sequencing libraries were generated using NEBNext^®^ Ultra^TM^ RNA Library Prep Kit for Illumina^®^ (NEB, Ipswich, MA, USA) following the manufacturer’s recommendations. Index codes were added to attribute sequences by each RNA sequencing method. Ultimately, PCR products of preferentially 250~300 bp in length were sequenced on an Illumina platform to generate 150 bp paired-end reads. Fastp v 0.19.3 [44] was used to filter the original data and to remove the low-quality reads. All subsequent analyses are based on clean reads. The reference genome of B73 v5.0 was downloaded from the National Center for Biotechnology Information (NCBI). HISAT v2.1.0 [45] and StringTie v1.3.4d [46] were used to compare clean reads to the reference genome to predict new genes. FeatureCounts v1.6.2 [47] was used to calculate the gene alignment and FPKM. DESeq2 v1.22.1 [48,49] was used to analyze differential expression between CK12h vs. HT12h and CK24h vs. HT24h, and the *p*-value was corrected using the Benjamini–Hochberg method. The corrected *p*-value and |Log_2_FC| or VIP value were used to set the threshold for a significant difference in gene expression. Enrichment analysis was performed based on the hypergeometric test.

### 4.3. Detection of Metabolites by Using UPLC-ESI-MS/MS

Three biological samples for each time point are freeze-dried with a vacuum freeze-dryer Scientz-100F (SCIENTZ, Ningbo, China). The 15 freeze-dried samples were crushed using a mixer mill MM 400 (RETSCH, Haan, Germany) with a zirconia bead for 1.5 min at 30 Hz. Lyophilized powder (100 mg) was dissolved in 1.2 mL 70% methanol solution, vortexed 30 s every 30 min for six times in total, and stored in a refrigerator at 4 °C overnight before the samples were centrifuged at 13,000× *g* for 10 min, the extracts were filtered through 0.22 μm SCAA-104 (ANPEL, Shanghai, China), then analyzed by using an ultra-performance liquid chromatography–electrospray ionization tandem mass spectrometry (UPLC-ESI-MS/MS) system (UPLC, SHIMADZU Nexera X2, Kyoto, Japan; MS, Applied Biosystems 4500 Q TRAP, Waltham, MA, USA).

Measurements were performed by a gradient program and the effluent was alternatively connected to an ESI-triple quadrupole-linear ion trap (QTRAP)-MS. LIT and triple quadrupole (QQQ) scans were acquired on a triple quadrupole-linear ion trap mass spectrometer (Q TRAP), AB4500 Q TRAP UPLC/MS/MS System, equipped with an ESI Turbo ion–spray interface, operating in positive and negative ion mode and controlled by Analyst 1.6.3 software (AB Sciex, Redwood, CA, USA). Instrument tuning and mass calibration were performed with 10 and 100 μM polypropylene glycol solutions in QQQ and LIT modes, respectively. QQQ scans were acquired as MRM experiments with collision gas (nitrogen) set to medium. DP and CE for individual MRM transitions were done after further optimization. A specific set of MRM transitions were monitored for each period according to the metabolites eluted within this period.

### 4.4. Measurement of Chlorophyll Fluorescence Parameters

The chlorophyll fluorescence parameters were measured by using PAM-2500 (WALZ, Effeltrich, Germany). Samples were moved to a growth chamber without light for 30 min, the saturation pulse (2800 μM·m^−2^ s^−1^) treatment was given to the leaf after sufficient dark adaptation, and then the values of chlorophyll fluorescence parameters, such as Fv/Fm, Y(II), qP, ETRmax, E_K_, Y(NPQ), were calculated by PAM-2500 with selected mode according to the manufacturer’s standard protocol.

### 4.5. Weighted Gene Co-Expression Network Analysis

WGCNA [50] was performed to investigate the relationship between DEGs and the metabolites detected. The power estimate was 18, the minimum module size was 50, and the merge cut height was 0.25.

### 4.6. Real-Time RT-qPCR for DEGs

The quality and concentration of extracted RNA from 15 samples were determined on the Agilent Bioanalyzer 2100 system (Agilent, CA, USA), treated with PrimeScript^TM^ RT Reagent Kit with gDNA Eraser (TAKARA, Dalian, China) according to the manufacturer’s standard protocol. RT-qPCR was conducted with ChamQ Universal SYBP qPCR Mix (VAZYME, Nanjing, China) on a QuantStudio^TM^ 5 Real-Time PCR system (Applied Biosystems, Foster, USA). *ZmActin1* (ID: 100282267) was used as a reference gene. Primers were designed by using PrimerBLAST on the website https://www.ncbi.nlm.nih.gov/tools/primer-blast (accessed on 30 January 2023). Primer sequences are listed in Appendix A. The relative level of gene expression was calculated by the 2^−ΔΔCt^ method according to the Ct value. Results were expressed as mean ± SE (*n* = 3).

### 4.7. Statistical Analysis

Significance tests were determined by one-way ANOVA analysis using SPSS v19 (IBM Corp., Armonk, NY, USA). All the qualifications were detected with three replicates, and results were performed as mean ± SD/SE. Heatmap and enrichment bubble plot were plotted online via the website https://cloud.metware.cn. Venn diagram was plotted following Bardou’s method [51] online via the website http://www.bioinformatics.com.cn.

## 5. Conclusions

To understand the regulation mechanisms of sweet corn encountering high temperatures, we performed transcriptomic and metabolomic analyses at the trefoil stage. When exposed to stress from elevated temperatures, sweet corn maintains growth by regulating gene expression in various pathways, especially the phenylpropanoid biosynthesis and its downstream pathways, such as monolignol biosynthesis, scopolin biosynthesis, coumarin biosynthesis, and flavonoid biosynthesis, with most of them being inhibited, showing that heat stress-induced changes in cell wall morphology play a vital role in plant thermotolerance.

Heat stress promoted the accumulation of scopoletin-7-o-glucoside (scopolin), 5,7-dihydroxy-3′,4′,5′-trimethoxyflavone, and 5,6,7,4′-tetramethoxyflavanone up to 24 h or longer, indicating a potential role of those compounds in heat response. Additionally, the inhibition of photosynthesis may be another efficient strategy to protect the photosystem under heat stress. Integrating analysis between genes and metabolites showed that the arginine biosynthesis, glutathione metabolism, tryptophan biosynthesis, endoplasmic reticulum, ribosome, endocytosis, and biosynthesis of cofactors pathways might also participate in sweet corn heat tolerance, and the hub genes of *RDR2*, *UGT73C5*, *LOC103633555*, and *CID7* seemed to be potential resistance markers. In conclusion, our results provide a new understanding of sweet corn heat resistance and constitute useful gene resources for thermotolerance crop breeding.

## Figures and Tables

**Figure 1 ijms-24-10845-f001:**
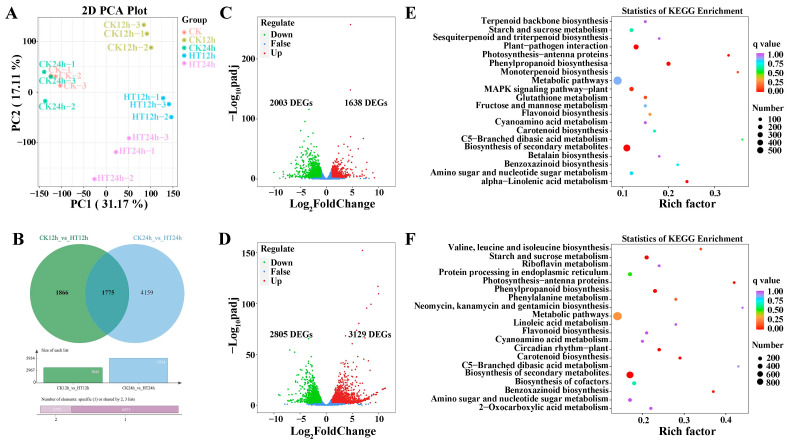
Transcriptome analysis for sweet corn in response to heat stress. PCA analysis among different time points under heat stress (**A**). Veen diagram between two groups at 12 h and 24 h under heat stress (**B**). Volcano plot shows DEGs with the conditions |Log_2_FC| ≥ 1 and FDR < 0.05 under heat stress at 12 h (**C**) and 24 h (**D**) under heat stress. KEGG enrichment analysis for sweet corn after 12 h (**E**) and 24 h (**F**) under heat stress.

**Figure 2 ijms-24-10845-f002:**
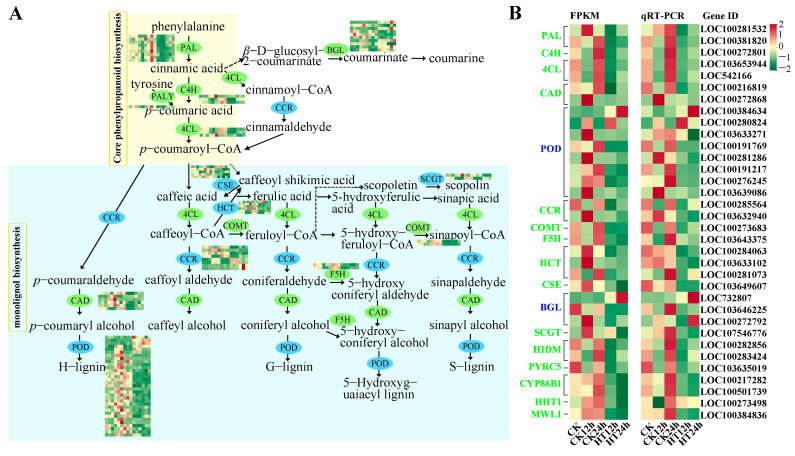
DEGs involved in phenylpropanoid biosynthesis and the phenylpropanoid metabolic process in response to heat stress. A graph of DEGs in phenylpropanoid biosynthesis and phenylpropanoid metabolic process under heat stress (**A**). Compare gene expression patterns between RNA-seq files (FRKM) and RT-qPCR data (**B**). The colors of ovals and letters in (**A**,**B**) are green and blue, respectively, indicating down-regulation of gene expression and simultaneous up- and down-regulation.

**Figure 3 ijms-24-10845-f003:**
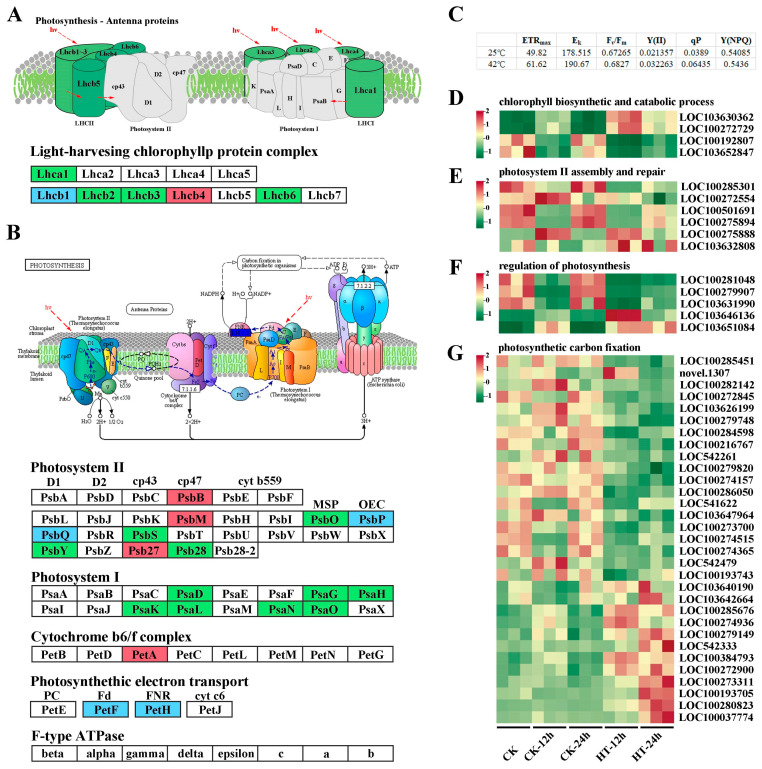
DEGs involved in the photosynthesis process in response to heat stress. Diagram of DEGs in photosynthesis–antenna proteins and photosynthesis process under heat stress. Red, green, and blue boxes represent up-, down-, and both up- and down-regulated gene expression, respectively (**A**,**B**). Index values after 24 h recovery for 12 and 24 h heat treatment seedlings (**C**). Heatmap of genes related to chlorophyll biosynthetic and catabolic (**D**), photosystem II assembly and repair (**E**), regulation of photosynthesis (**F**), and photosynthetic carbon fixation (**G**). ETRmax, maximum electron transport rate; E_K_, saturating irradiance; F_v_/F_m_, maximum photosynthetic rate; Y(II), actual photosynthetic rate; qP, photochemical quenching coefficient; and Y(NPQ), quantum yield of regulatory energy dissipation.

**Figure 4 ijms-24-10845-f004:**
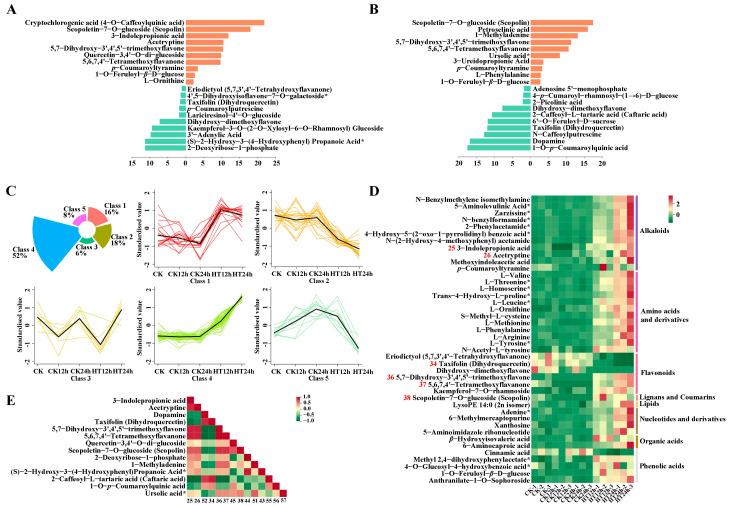
Metabolomic profiles of sweet corn seedlings under heat stress. Top 20 metabolites detected under heat stress after 12 h (**A**) and 24 h (**B**). Orange bars stand for up-regulated compounds, and cyan bars stand for down-regulated compounds. Select conditions were VIP ≥ 1 and |Log_2_FC| ≥ 1. Cluster results of 130 metabolites into five groups at different time points (**C**). A heatmap for 42 metabolites detected in the groups CK12h vs. HT12h and CK24h vs. HT24h (**D**). Spearman correlation results among 14 metabolites (**E**). The serial number in D (red color) and E (dark color) means corresponding metabolites in Appendix A. According to the principle of mass spectrometry detection, the metabolites with isomers cannot be distinguished should marked with “*”, as shown in (**A**,**B**,**D**,**E**).

**Figure 5 ijms-24-10845-f005:**
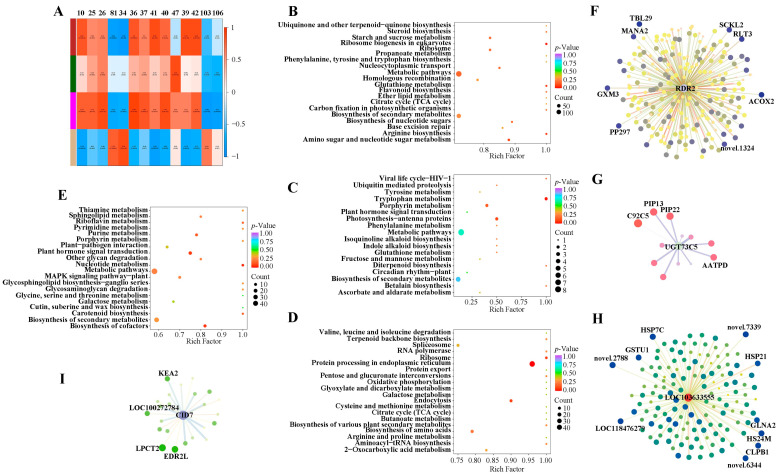
Correlation analysis to mine key pathways and hub genes involved in heat response in sweet corn. Spearman correlation analysis between metabolites and DEGs. The squares with different colors in the first column presented different modules (**A**). KEGG enrichment analysis of brown (**B**), dark green (**C**), magenta (**D**), and tan (**E**) modules. Hub genes identified in the brown (**F**), dark green (**G**), magenta (**H**), and tan (**I**) modules. Log_2_FC of each connected gene is shown by the thickness of lines and the size of circles; the degree between the hub gene and connected genes is indicated by the color intensity of the circle color. The most connected genes are marked. Abbreviations for genes: *RDR2*: *RNA*-*dependent RNA polymerase 2*; *UGT73C5*: *UDP*-*glycosyltransferase 73C5*; *CID7*: *CTC*-*interacting domain 7*; *ACOX2*: *acyl*-*coenzyme A oxidase 2*; *TBL29*: *trichome birefringence*-*like 29*; *RLT3*: *ringlet 3*; *SCKL2*: *fructokinase*-*like 2*; *PP297*: *pentatricopeptide repeat*-*containing protein*; *GXM3*: *glucuronoxylan 4*-*O*-*methyltransferase 3*; *MANA2*: *probable alpha*-*mannosidase*; *C92C5*: *cytochrome P450 92C5*; *PIP13*: *plasma membrane intrinsic protein 1*-*3*; *PIP22*: *plasma membrane intrinsic protein 2*-*2*; *AATPD*: *AAA*-*ATPase*; *HS24M*: *24.1 kDa heat shock protein*; *CLPB1*: *casein lytic proteinase B1*; *GSTU1*: *glutathione S*-*transferase 1*; *GLNA2*: *glutamine synthetase root isozyme 2*; *HSP21*: *heat shock protein 21*; *HSP7C*: *heat shock cognate 70 kDa protein*; *LPCT2*: *lysophospholipid acyltransferase 2*; *EDR2L*: *enhanced disease resistance 2*-*like*; *KEA2*: *K(+) efflux antiporter 2*.

## Data Availability

Raw data have been deposited in the NCBI Short Read Archive (SRA) database, and the accession number is PRJNA933114.

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
