# Peer review of "The Regulatory Network of Sweet Corn (Zea mays L.) Seedlings under Heat Stress Revealed by Transcriptome and Metabolome Analysis"

_ijms, 2023, doi:10.3390/ijms241310845_

Round 1

Reviewer 1 Report

In the submitted manuscript, the authors described the regulatory network in sweet corn at the seedling stage based on the differentially expressed genes (DEGs) and pathways detected, integrated with the in vivo metabolic changes during heat stress. The aim of this study was to uncover the regulatory mechanism of sweet corn response to heat stress by 18 integrating transcriptome and metabolome analyses. . A regulatory network was built by analyzing the correlations between gene modules and metabolites, and four hub genes in sweet corn seedlings under heat stress were identified: RNA-dependent RNA polymerase 2 (RDR2), UDP-glucosyltransferase 73C5 (UGT73C5), LOC103633555, and CTC-interacting domain 7 (CID7). These results provide a foundation for improving sweet corn development through biological intervention or genome-level modulation.

The authors conducted extensively in silico analyses of the studied genes, combined with the experimental validation of gene expression in transcript level by quantitative real-time RT-PCR. The strong points of the manuscript is a very interesting topic and application of NGS technology. The description of identified transcripts is very interesting. However, I have some questions about the methodology.

The current guidelines for expressional analysis are that experiments using RNA - Seq data to describe changes should contain the biological replicates (unless otherwise justified) herein authors did not specify how many replicas were used.

Each biological replicate should be represented in an independent library, each with unique barcodes, if libraries are multiplexed for sequencing. In this case, authors did not specify this tasks.

Authors performed validation for DEGs by quantitative RT-PCR. Actually, the information given by the author in the manuscript about the replicas is also not clear. According to the standard protocol, it  must contain three biological and three technical replicates. Herein there is lack of this information and thus it seems the qPCR analysis are not properly replicated.

In addition, the authors only mention that the qPCR results overlap with the RNA_seq results. This should bepresented in one Figure.

What about the primers for qPCR techniques? Authors did not mention about software for primers design. 

Equally important is to explain how many replicates were used in the analysis of the metabolites in question. Here usually more replicates are used, but how many the authors used and whether the study is reliable is unknown.

The above comments are substantive and the quality of the work depends on it.

good

Author Response

Thank you very much for positive comments on our manuscript and great suggestions that are very helpful to improve our manuscript.

Comment 1) The current guidelines for expressional analysis are that experiments using RNA - Seq data to describe changes should contain the biological replicates (unless otherwise justified) herein authors did not specify how many replicas were used.

Response for comment 1)

Thank you for this comment. In our study, samples were taken after 12 h and 24 h under 42°C heat treatment, parallel control for each time points under 25°C were also taken, 0 h as blank control. Three replications were set for each time point. We totally have 15 samples for the construction of RNA-Seq libraries. We introduce this in “4.1. Growing conditions and stress for sweet corn seedings” in line 529-534.

Comment 2) Each biological replicate should be represented in an independent library, each with unique barcodes, if libraries are multiplexed for sequencing. In this case, authors did not specify this tasks.

Response for comment 2)

Thank you for this comment. In our study, each biological replicate represented an independent library, libraries did not been multiplexed for sequencing. Raw data have been deposited in the NCBI Short Read Archive (SRA) database, and the accession number is PRJNA933114 (line 638). We have unique barcodes for the 15 libraries, the accession number from SRR23411761 to SRR23411775.

Comment 3) Authors performed validation for DEGs by quantitative RT-PCR. Actually, the information given by the author in the manuscript about the replicas is also not clear. According to the standard protocol, it must contain three biological and three technical replicates. Herein there is lack of this information and thus it seems the qPCR analysis are not properly replicated.

In addition, the authors only mention that the qPCR results overlap with the RNA_seq results. This should bepresented in one Figure.

Response for comment 3)

Thank you very much for this suggestion. We performed validation for DEGs by quantitative RT-qPCR, each time point has three biological replications and original data were shown in Table S6. In the table, we listed the time point with three biological replications, for example CK-1, -2, -3, each biological replication had three technical replications, which were shown by mean ± SE (n = 3).

We mentioned that the qPCR results overlap with the RNA-Seq results in Figure 2B with heatmaps, we use one square to stand for the average value for the three biological replication, for example CK stand for the mean value for CK-1, -2, -3. FPKM represents RNA-Seq results and qRT-PCR stand for qPCR results.

Comment 4) What about the primers for qPCR techniques? Authors did not mention about software for primers design.

Response for comment 4)

Thank you for this comment. We use PrimerBLAST to design unique primers for each gene. And we add this and the website in the methods (line 594-595).

Comment 5) Equally important is to explain how many replicates were used in the analysis of the metabolites in question. Here usually more replicates are used, but how many the authors used and whether the study is reliable is unknown.

Response for comment 5)

Thank you for this good suggestion. Three biological replications were used for metabolites detection, we add the explain in line 555-556. The original three biological replication data for detected metabolites were listed in Table S8.

Reviewer 2 Report

The submitted manuscript deals with the current issue of the effect of high temperatures on plants. For this reason, I believe that the mentioned research is up-to-date and at the same time the obtained results have considerable potential not only for the further direction of research. The issue of plant defense mechanisms based on genetic analyzes is very complicated with regard to the heterogeneity of the plant body. The manuscript is written relatively carefully with logical continuity. Perhaps it would be appropriate to add more information about heat shock proteins in the introductory section. The results are based on the description of graphs and diagrams. It's a shame that the pictures and graphs are often small, making them somewhat harder to read. I recommend, if possible, to make them bigger. The chlorophyll fluorescence method is mentioned in the methodology, but in the results section these parameters are described only in general terms. I would divide the discussion into the same subsections as the results, which would make it easier for the reader to navigate the text. It is descriptive in places. The methodology is laid out clearly and comprehensibly. Nevertheless, I recommend supplementing the nutrient content and irrigation in the case of plant cultivation. What sheets were taken? Are these photosynthetically mature leaves? Please check the citations to see if all cited journals are written in international abbreviations.  

Author Response

We gratefully appreciate positive comments and considerate suggestions for our manuscript.

Comment 1) Perhaps it would be appropriate to add more information about heat shock proteins in the introductory section.

Response for comment 1)

Thank you very much for this great suggestion. We have added more information about heat shock proteins in the introductory section in line 60-69.

Comment 2) It's a shame that the pictures and graphs are often small, making them somewhat harder to read. I recommend, if possible, to make them bigger.

Response for comment 2)

We agree that we should try to show much bigger pictures and graphs. We have rearrangement the images and enlarge the words to make them easier to read. We hope this new Figures are acceptable. Details are shown in the manuscript (line 133, 205, 319, 368).

Comment 3) The chlorophyll fluorescence method is mentioned in the methodology, but in the results section these parameters are described only in general terms.

Response for comment 3)

Thank you for this good suggestion. We have added the detected values to make it easier to understand the results section. Details are shown in line 222-226.

Comment 4) I would divide the discussion into the same subsections as the results, which would make it easier for the reader to navigate the text.

Response for comment 4)

Thank you for this good suggestion, we have merged some paragraphs as the results to make it easier to read, such as paragraph 4 and 5, 6 and 7, 8 and 9. Details are shown in the manuscript (line 468, 484, 503).

Comment 5) Nevertheless, I recommend supplementing the nutrient content and irrigation in the case of plant cultivation. What sheets were taken? Are these photosynthetically mature leaves?

Response for comment 5)

We agree that it is necessary to exhibition the culture conditions, so we add the details as “Germinated seeds were cultivated with adequate irrigation and given water-soluble compound fertilizer once a week. After 21 days growth to trefoil stage, the fully extended and photosynthetically mature second leaves from five seedlings were taken for each sample after 42°C for 12 h and 24 h” in line 529-531.

Comment 6) Please check the citations to see if all cited journals are written in international abbreviations.

Response for comment 6)

Thank you for this good suggestion, we have checked all the citations and make sure all cited journals are written in international abbreviations, details are added in the References part (line 644, 645, 664, 669, 673, 682, 684, 688, 705, 706).

Round 2

Reviewer 1 Report

The authors answered my questions.